# Evaluation of the Foot Center of Pressure Estimation from Pressure Insoles during Sidestep Cuts, Runs and Walks

**DOI:** 10.3390/s22155628

**Published:** 2022-07-27

**Authors:** Pauline Morin, Antoine Muller, Charles Pontonnier, Georges Dumont

**Affiliations:** 1University Rennes, CNRS, Inria, IRISA-UMR 6074, 35000 Rennes, France; charles.pontonnier@ens-rennes.fr (C.P.); georges.dumont@ens-rennes.fr (G.D.); 2University Lyon, University Gustave Eiffel, University Claude Bernard Lyon 1, LBMC UMR_T 9406, 69622 Lyon, France; antoine.muller@univ-lyon1.fr

**Keywords:** biomechanics, motion analysis, ground reaction forces, outside laboratory experiments, moticon

## Abstract

Estimating the foot center of pressure (CoP) position by pressure insoles appears to be an interesting technical solution to perform motion analysis beyond the force platforms surface area. The aim of this study was to estimate the CoP position from Moticon${}^{®}$ pressure insoles during sidestep cuts, runs and walks. The CoP positions assessed from force platform data and from pressure insole data were compared. One calibration trial performed on the force platforms was used to localize the insoles in the reference coordinate system. The most accurate results were obtained when the motion performed during the calibration trial was similar to the motion under study. In such a case, mean accuracy of CoP position have been evaluated to 15±4mm along anteroposterior (AP) axis and 8.5±3mm along mediolateral (ML) axis for sidestep cuts, 18±5mm along AP axis and 7.3±4mm along ML axis for runs, 15±6mm along AP axis and 6.6±3mm along ML axis for walks. The accuracy of the CoP position assesment from pressure insole data increased with the vertical force applied to the pressure insole and with the number of pressure cells involved.

## 1. Introduction

The study of human movement outside the laboratory is a major application issue for many sectors (sports, ergonomics, monitoring of physical activity, etc.). The center of pressure (CoP) position appears to be an interesting biomechanical quantity in the analysis of movement. This quantity can be directly used to evaluate balance or reactivity in sports performance [1,2,3,4]. It can also be used as an intermediate quantity to describe the foot–ground contact [5]. It can be used as a driving quantity to estimate the reaction forces between the subject and the ground using inverse dynamics methods [6].

Force platforms are considered the gold standard for CoP position assessment [7,8,9,10]. These devices reduce the ecological validity of the experiments while limiting the area of motion to the dimensions of the force platform [11]. Estimating the CoP position using pressure insoles removes this limitation by providing more ecological experimental conditions at the cost of a limited impact on subject motion [12]. The evaluation of CoP position assesment from pressure insole data (piCoP) opens the door to applications where the magnitude of the studied phenomenon is greater than the accuracy determined [13].

Several studies have evaluated the reliability and repetability of the piCoP position for different pressure insole models. Moticon${}^{®}$ insoles had strong reliability and demonstrated lower validity compared to Pedar-X${}^{®}$ insoles [14]. A dependence of the piCoP position accuracy to the vertical force applied to the insole has been demonstrated [15] by comparing piCoP to CoP position measured with the force platforms (fpCoP) for leg press and squat exercises.

The accuracy of the CoP position evolution estimated by means of pressure insoles was evaluated on gait [7,8,9]. Results on gait cannot be extended as results valid in a sport environment, which is characterized by the variety and the dynamics of sport movements. During sidestep cuts, acceleration along mediolateral axis causes the trajectory to change. In contrast to straight-line movements, sidestep cuts are characterized by a significant shift of the CoP along mediolateral axis [16].

In view of the use of pressure insoles in the inverse dynamics process, an evaluation of the estimation of the ground reaction force and the estimation of the CoP position is necessary. Recent studies presented methods to estimate ground reaction force from plantar pressure, considering CoP as the force application point [17]. The artificial neural network use involves a large amount of data [18].

The purpose of this study was to evaluate the accuracy of the CoP position evolution estimated by means of Moticon${}^{®}$ pressure insoles during sidestep cuts, runs and walks. The piCoP and fpCoP were compared after synchronization and insole localization in the reference coordinate system. The accuracy of the piCoP was studied as a dependent variable of the kind of the motion performed during the calibration trial, the vertical external forces estimated by pressure insoles and the number of pressure cells involved in the piCoP computation considered as independent variables.

## 2. Materials and Methods

### 2.1. Experimental Procedure

The cohort was composed of 4 female and 10 male subjects (age: 29 ± 2 years old, height: 1.8 ± 0.1 m, mass: 70 ± 10 kg). The inclusion criterion was to wear EU size 38, 39, 42 or 43 shoes, since two insole sizes were available (38–39 and 42–43 EU). Each subject signed an informed consent form and a pseudonimization protocol was followed for data storage. The experimental protocol was approved by the INRIA National Ethics Committee (Comité Opérationnel d’Evaluation des Risques Légaux et Ethiques, 2021-06, 02/22/2021).

Whole body motion capture was performed by placing 46 reflective markers on standardized anatomical landmarks according to the recommendations of the International Society of Biomechanics [19,20]. Only markers placed on the feet were used for this study; the full body kinematics were measured for further studies. Motion capture data were recorded with a Qualisys optoelectronic motion capture system (200 Hz, 22 “12 Mpixels OQUS 7+” cameras). Ground reaction forces and moments (GRF&M) were recorded with two AMTI force platforms (2000 Hz). Underfoot pressure, CoP position, vertical force and feet acceleration were recorded by Moticon${}^{®}$ pressure insoles (OpenGo, 100 Hz, 16 pressure cells covering 65% of the insole area with a 0.25 $N\cdot {\mathrm{cm}}^{-2}$ resolution and a hysteresis <1% [21], 1 inertial measurement unit (IMU) at the center of each insole). The data from the insoles were linearly interpolated from 100 Hz to 200 Hz to match with the Qualisys system frequence. The wireless insoles were inserted into the subjects’ personal sport shoes.

Prior to data collection, the pressure insoles were reset and calibrated according to the manufacturer’s recommendations. During the experiment, each subject performed five sidestep cuts, five runs and five walks at comfort speed (15 trials per subject to achieve acceptable statistical comparisons [22]). The sidestep cuts were always performed using the right foot as the support foot. Subjects were instructed to modulate their strides so as not to touch both platforms during a given step. In total, 10 trials out of 210 were removed from this study because of pressure insole malfunction.

### 2.2. Method Overview

Three steps were developed to perform the piCoP and the fpCoP position comparison (Figure 1):Synchronization and frame selection: all data were synchronized, and when piCop and fpCoP data were available for comparison, frames were selected;Localization: the piCoP and fpCoP were expressed in the same reference coordinate system;Comparison: the fpCoP and the piCoP were statistically compared.

### 2.3. Synchronisation and Frame Selection

For the motion capture data, for each trial, an operator manually, and by visual control, selected frames where both data were available for comparison, as shown in Figure 2c (feet not outside the force platforms, as shown in Figure 2a, and not on the same platform, as shown in Figure 2b).

To select the corresponding frames in the insole data, the foot acceleration computed from the motion capture data (barycenter of the markers placed on the shoe shown in Figure 3) and the acceleration provided by the IMU insole data were cross-correlated. The cross-correlation between the two accelerations enabled to detect the motion cycle executed on the force platforms in the insole data. To synchronize the selected motion cycle in the motion capture data and the selected motion cycle in the insole data, the vertical ground reaction forces measured from the force platforms and pressure insoles were cross-correlated.

Frames with foot–ground contact were selected from the synchronized data. Foot–ground contact was considered when contact was simultaneously detected by the pressure insole and by the force platform. Contact on the pressure insole was considered as active when at least two pressure cells were activated. A pressure cell was considered as active when the measured pressure was greater than 1.5 $N\cdot {\mathrm{cm}}^{-2}$, excluding low-intensity pressure peaks (corresponding to the pressure measurement noise, defined empirically as peaks of intensity less than 2 $N\cdot {\mathrm{cm}}^{-2}$ and duration less than 0.03 s). Contact between the foot and the force platform was considered as active when the force platform measured a vertical force greater than 75 N (empirical value).

### 2.4. Localization

The motion capture reference frame was the optoelectronic reference coordinate system ($R_{0}$). Force platform data were expressed in $R_{0}$. The piCoP position expressed in the insole coordinate sytem (${}^{\mathrm{pi}}$piCoP) required localization in $R_{0}$ for comparison. The transformation matrix from the pressure insole reference frame to $R_{0}$, ${}^{0}T_{\mathrm{pi}}$ was decomposed into two successive transformation matrices:–Transformation matrix from the pressure insole reference frame to the foot reference frame ${}^{f}T_{\mathrm{pi}}$;–Transformation matrix from the foot reference frame to $R_{0}$: ${}^{0}T_{f}\left(t\right)$.

The matrix ${}^{0}T_{f}\left(t\right)$ was built from the markers position and the foot reference frame shown in Figure 3.

The matrix ${}^{f}T_{\mathrm{pi}}$ was considered as constant assuming small relative displacements between the insole, the foot and the markers. ${}^{f}T_{\mathrm{pi}}$ was estimated by comparing the successive fpCoP position expressed in $R_{0}$ (${}^{0}$fpCoP) and the successive piCoP position expressed in $R_{0}$ (${}^{0}$piCoP) during a valid sequence of foot–ground contact frames for each foot. The contact sequence was selected from one trial designed as calibration trial. ${}^{f}T_{\mathrm{pi}}$ was determined to minimize the root mean square error (RMSE) between ${}^{0}$piCoP and ${}^{0}$fpCoP (Figure 4) [8,9].

${}^{0}T_{f}\left(t\right)$ was approximated as constant during the contact. The successive positions of the CoP were contained in the successive contact areas between the foot and the platform. These contact areas resulted from the deformation of the shoe on the ground (the motion was assumed to be slip-free). The overview of those surfaces was similar to the contact surface when the entire foot was flat on the floor at the frame designed as t = $t_{f}$. The successive reference frames of the contact surfaces were approximated by the reference frame when the entire foot was flat on the ground, ${}^{0}T_{f}\left(t_{f}\right)$. For each foot, ${}^{0}T_{f}\left(t_{f}\right)$ was computed as ${}^{0}T_{f}\left(t\right)$ when the maximum of pressure cell detect pressure at $t=t_{f}$.

### 2.5. Comparison

The accuracy of piCoP was assessed by the RMSE between ${}^{0}$piCoP and ${}^{0}$fpCoP along the anteroposterior (AP) and mediolateral (ML) axes in $R_{0}$. For each trial, RMSEs were computed successively using each trial performed by the same subject as the calibration trial (15 × 15 RMSE were computed for each subject).

The impact of the kind of motion executed during the calibration trial on the piCoP position accuracy was studied. The motion executed during the studied trial was named the trial motion (TM) (sidestep cuts, run or walk) and the motion executed during the calibration trial was named the calibration trial motion (CTM). Regardless of the RMSE obtained by using as calibration trial the studied trial and for each TM value, CTM were tested as significant variables on RMSE (AP and ML component). For each pair of values (TM; CTM), the normality distribution of the RMSEs were tested with a Shapiro–Wilk test (3 × 3 tests). For each pair of values (TM; CTM), the different results were considered as repeated measures. For each TM, the influence of the CTM was tested with a Friedman test. Pairwise comparison completed the statistic study with a Durbin–Conover test.

The impact of the vertical force intensity and the impact of the number of activated pressure cells on the piCoP accuracy was studied. At each frame, the piCoP absolute error, the intensity of the vertical force captured by the pressure insole and the number of activated cells were recorded. Focusing on the results obtained using the studied trial as a calibration trial, the dependence between the piCoP absolute error and the vertical force applied on the insole or the number of activated cells was studied. The vertical force was grouped to fit 100 N force intervals. The piCoP absolute errors were plotted as a function of the 100 N vertical force intervals. The dependence between the piCoP absolute error and the number of activated cells was tested with a Friedman test.

## 3. Results

The average movement speed was $3.3\;\pm \;0.4\;m\cdot s^{-1}$ for sidestep cuts, $3.9\;\pm \;0.4\;m\cdot s^{-1}$ for runs and $1.6\;\pm \;0.2\;\;m\cdot s^{-1}$ for walks. The range of the piCoP shift was evaluated on average to be 137mm along the AP axis and 50.7mm along the ML axis for sidestep cuts, 150mm along the AP axis and 24.4mm along the ML axis for runs and 176mm along the AP axis and 39.5mm along the ML axis for walks.

Regardless of the RMSE obtained by using the studied trial as the calibration trial, the RMSE distribution is shown in Figure 5 for each (TM; CTM) value. For sidestep cuts trials, the RMSE distribution is represented on the left part in Figure 5a (RMSE along AP axis) and in Figure 5a (RMSE along ML axis). The boxplot color correspond to the CTM. Regardless of the CTM, the RMSE was 15±4mm along the AP axis and 8.5±3mm along the ML axis for sidestep cuts, 18±5mm along the AP axis and 7.3±4mm along the ML axis for runs and 15±6mm along the AP axis and 6.6±3mm along the ML axis for walks.

According to a Friedman test, the difference between the RMSE distribution according to the CTM was statistically significant (p< 0.001) for each TM value, except along the AP axis for walk trials (on the right part in Figure 5a). Along the ML axis, mean RMSEs were consistently lower when CTM and TM were the same. Those pairwise comparison are statistically signifiant according to the Durbin–Conover test.

Focusing on the results obtained by using the studied trial as the calibration trial, the mean RMSE was 13 ± 4 mm (9.5% of the mean piCoP shift along AP axis) along the AP axis and 7.4 ± 3 mm (14% of the mean piCoP shift along ML axis) along the ML axis for sidestep cuts, 16 ± 5 mm (11%) along the AP axis and 5.3 ± 2 mm (21%) along the ML axis for runs and 12 ± 3 mm (6.8%) along the AP axis and 4.2 ± 1 mm (13%) along the ML axis for walks.

For each frame, the piCoP absolute error is plotted in Figure 6 as a function of the vertical force (grouped in 100 N intervals) captured by the pressure insole. For each force interval, the mean number of activated pressure cells is plotted. The mean number of activated cells was between 2 (the criterion for contact detection was to detect at least 2 activated cells) and 16 (number of cells on each insole). There were frames where all pressure cells are activated. According to a Friedman test, the dependence of the piCoP absolute error distribution to the number of activated cells was statistically significant (p<0.001) for the AP axis and for the ML axis. The accuracy of the piCoP increased with the number of activated cells and with the vertical force captured by the pressure insole.

## 4. Discussion

CoP positions estimated by the Moticon${}^{®}$ pressure insoles were evaluated for sidestep cuts, runs and walks. Accuracy depended on the vertical force applied to the insole and the number of activated pressure cells.

### 4.1. Pressure Insole Accuracy

The piCoP assessment RMSE were consistent with the literature. Focusing on the walk trials results obtained using the studied trial as the calibration trial, the method accuracy was similar to other studies [9,23]. A similar study evaluated RMSE on gait to 43 ± 13 mm for the AP component and 6.3 ± 3 mm for the ML component [9]. One study [15] evaluated the RMSE to be 12 mm along the AP axis and 3.9 mm along the ML axis for the squat and 12 mm along the AP axis and 4.5 mm along the ML axis for the leg press exercise. The differences in motions and insole models (Pedar-X${}^{®}$ in [15]) may explain these more accurate results than in the present study. The Moticon${}^{®}$ insoles showed lower validity compared to Pedar-X${}^{®}$ insoles [14].

The CTM had an impact on the accuracy of the piCoP estimated (Figure 5). The studied motions had different GRF&M characteristics in term of range, frequency and component preponderance. The study confirms that sidestep cuts admit a larger range of CoP shift along ML axis than runs and walks [16]. Runs and sidestep cuts were more dynamic and admitted larger range of vertical forces than walks. Therefore, it is recommended that the CTM and TM are similar to maximize the accuracy of CoP estimation in the experimental protocols.

The accuracy of the piCoP estimation increased with the number of activated cells and with the vertical force applied on the insole (Figure 6). The greater the number of activated cells involved in the CoP estimation, the less impact a pressure cell error had on the estimation. The greater the vertical force applied to the insole, the less impact the noise from the pressure measurement had on the estimation. For high values of vertical force applied on the insole, the mean accuracy of the piCoP estimation along the ML axis, increased and the maximum error increased. These high values of vertical force appeared during impact phases of dynamic movements (when the foot hits the ground during sidestep cuts and runs). Those errors may be explained by a significant deformation of the shoe and the insole or by the fact that the foot slips inside the shoe along the ML axis, especially during sidestep cuts.

The studied motions (sidestep cut, run and walk) and the diversity of subjects mass resulted in a larger range of vertical force values compared to a previous study [15]. The highest values in the present study were achieved during dynamic motions (sidestep cut and run) by the heaviest subjects. This previous study demonstrated the dependence of the accuracy of piCoP on the vertical force applied to the insoles. The present study extends this result to a larger range of vertical force (0 N to 2500 N).

### 4.2. Perspectives of Use

The study evaluated the CoP position assessed from the Moticon${}^{®}$ insoles. The RMSE values obtained using the studied trial as the calibration trial can be considered as the accuracy of the pressure insoles to estimate the CoP position in the pressure insole reference frame. The Moticon${}^{®}$ pressure insoles can be used to study any ativity involving a range of CoP displacements larger than two centimeters. The confidence in the piCoP assesment depends on the number of active pressure cells and the vertical force applied to the insole. Depending on the application of piCoP assessment, a number of active pressure cells or a vertical force threshold can be used to enhance the reliability of the piCoP estimation. In the study, frames with low values of active pressure cells (and vertical force) corresponded to frames at the beginning and the end of the contact sequence (heel strike and toe-off).

Pressure insoles can be used with an optoelectronic system to locate the position of the CoP in $R_{0}$. This localization is a critical point [7,8,9]. By using a calibration trial different from the one studied, a two-phase experimental protocol could be proposed: a calibration step performed on a force platform followed by a capture trial without limitation to the force platform area. Previous studies have investigated the following experimental protocol: during calibration, the pressure insole was static on the force platforms and an operator pressed on each pressure cell with a stick [7] to locate the pressure insole in the foot (or shoe) reference frame. This method did not take into account the possible shoe deformations related to the presence of the foot in it. The accuracy of this method may decrease with soft shoes. The RMSEs admitted similar values for any TM. The piCoP estimation accuracy does not depend on the TM. The Moticon${}^{®}$ pressure insole can be used to study sidestep cuts, runs and walks with the same accuracy or any displacement, including some of those motions.

RMSEs along the ML axis were lower than along the AP axis. This study confirms that the piCoP estimation is more accurate along the AP axis [10]. The piCoP estimation normed by the average shift along the AP axis and shift along the ML axis of the piCoP trajectory is more accurate along the ML axis. The interpretation of the pressure insoles’ accuracy depends on the perspective of use. From the perspective of using it for CoP visualization of motion [24], the piCoP estimation error along the ML axis has more impact than the piCoP estimation error along the AP axis. As a method considering the CoP position as a driving quantity to estimate GRF&M [6], the piCoP estimation error along the ML axis has less impact than the piCoP estimation error along the AP axis.

### 4.3. Limitations and Perspectives

The impact of synchronization, especially for dynamic motions such as runs and sidestep cuts, must be taken into account. A hardware solution for synchronization between motion data and pressure insole data should be implemented.

The use of pressure insoles should be questioned due to the limited capture frequency (100 Hz) for studies of phenomena of less than 0.1 s, including peaks [25], such as impulses in sprint studies [26].

The results for high vertical forces values were limited. The vertical force interval 2300–2400 N included only one value. In this case, the statistical processing cannot be considered as representative. A larger range of vertical force value should be captured to test the pressure insole over their full operating range. This larger range can be obtained during more dynamic motions, such as sprints or jumps.

The studied displacements admitted the AP axis as the main displacement direction. The range of the piCoP shift along the AP axis was larger than the range of the piCoP shift along the ML axis for any motion. No studied displacements admitted the ML axis as the main displacement direction.

This study showed lower accuracy when a small number of cells were activated and when a small vertical force was applied. Frames where a small number of cells was activated coincided with frames when the vertical force was small. Further studies should investigate if the accuracy of the piCoP estimation is dependent on the time phases of the selected tasks. Further experiments should include specific motions to independently question the impact on the piCoP accuracy of the number of activated cells and the vertical force applied to the pressure insole (e.g., with an arabesque motion on the ball of the foot to study high values of vertical force and low numbers of activated cells effects on accuracy, or sitting with the feet flat on the floor to study how low values of vertical force and a high number of activated cells affect accuracy).

The accuracy of the piCoP positions depends on the accuracy of measurement of the insoles and the accuracy of the estimation of the related reference frame. The error of the estimation of the reference frame of the insoles is minimized by using as a calibration trial the trial under study, but this is not totally compensated. The markers placed on the subject measure the shoe movement rather than the foot movement. During foot rolling, the pressure insole and the shoe were deformed. The transformation matrix between the foot and the insole was assumed to be constant over time and did not take this source of error into account. Another source of error came from the foot model, considered as one rigid solid. A two-solids foot model and using two associated transformation matrices between the foot and the global reference frame should be investigated to reduce the foot location error [27].

## 5. Conclusions

This study evaluated the accuracy and relevance of the CoP position evolution estimated by means of Moticon${}^{®}$ pressure insoles during sidestep cuts, runs and walks. Pressure insoles can replace force platforms to measure CoP positions with an accuracy about 15mm along the AP axis and about 8mm along the ML axis. This accuracy increases with the vertical force applied to the pressure insole and with the number of pressure cells involved.

This study evaluated an upper limit of the pressure insoles accuracy in the pressure insole coordinate system. This accuracy depends on the calibration trial used for the localization of the insoles in the reference coordinate system. The most accurate results were obtained when the motion performed during the calibration trial was similar to the motion under study.

## Figures and Tables

**Figure 1 sensors-22-05628-f001:**
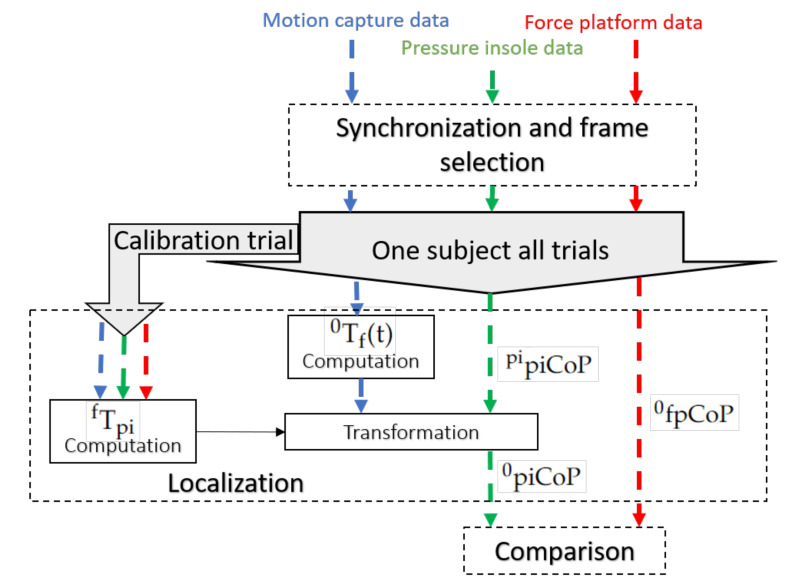
Method overview to evaluate the piCoP accuracy. This process used motion capture data (blue), pressure insole data (green) and force platform data (red). The localization used the processing of one calibration trial. Evaluation was based on the comparison between the fpCoP and the piCoP expressed in the reference coordinate system.

**Figure 2 sensors-22-05628-f002:**
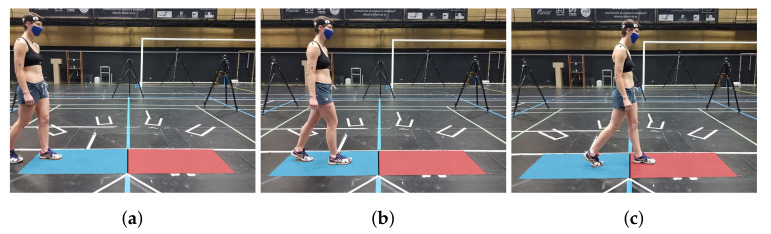
Photos of a walk trial: phase when the subject was out of the force platforms (**a**), when the subject was in the double support phase on the same platform (**b**) and when the subject had each foot on a separate platform (**c**). Phase (**a**,**b**) were rejected for the study. Phase (**c**) was selected for the study.

**Figure 3 sensors-22-05628-f003:**
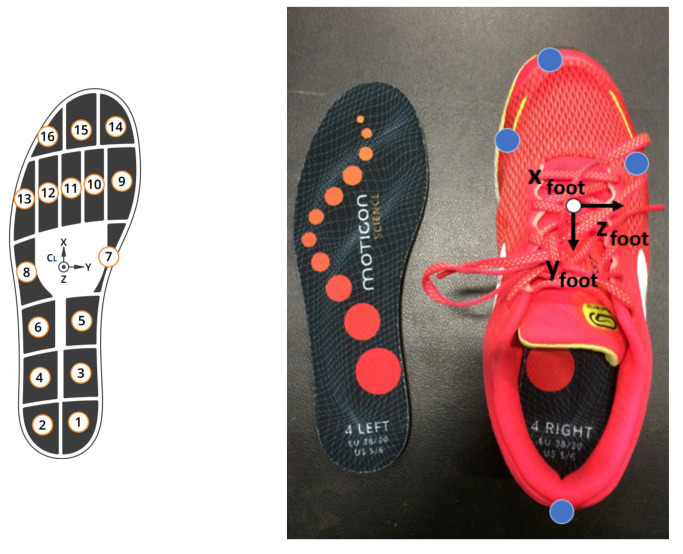
Shoe equipped with instrumented insoles and markers enabling the construction of the foot reference frame.

**Figure 4 sensors-22-05628-f004:**
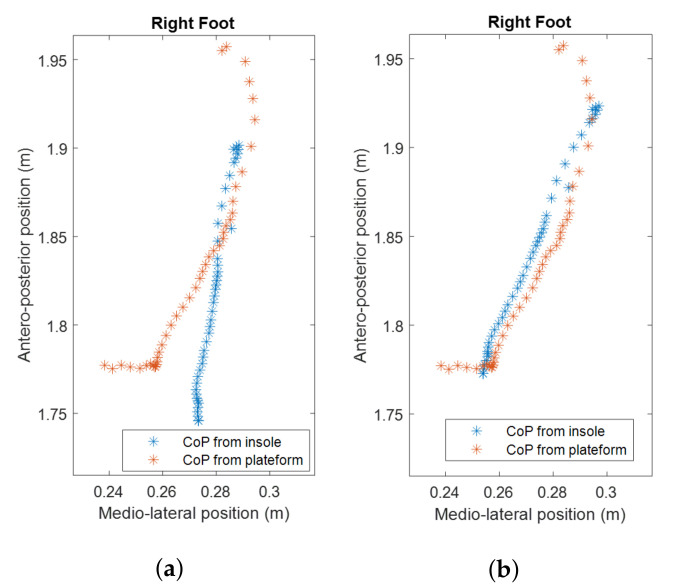
CoP trajectory comparison ${}^{f}T_{\mathrm{pi}}$ as identity matrix (**a**) and ${}^{f}T_{\mathrm{pi}}$ once calibrated (**b**) for a run trial (right foot).

**Figure 5 sensors-22-05628-f005:**
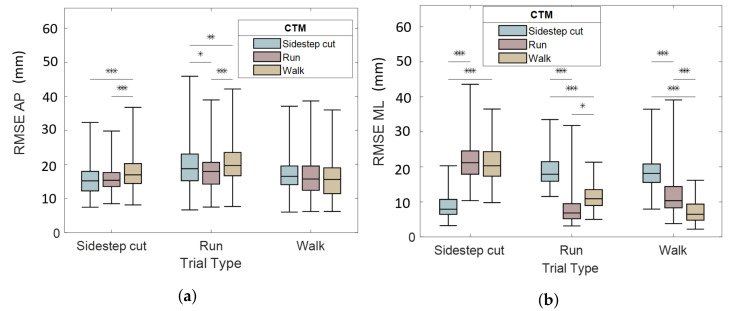
CoP position estimation RMSE between pressure insole and force platform: AP component (**a**) and ML component (**b**). Pairwise comparison (Durbin-Conover) is indicated with “***” for p<0.001, “**” for p<0.01, “*” for p<0.05.

**Figure 6 sensors-22-05628-f006:**
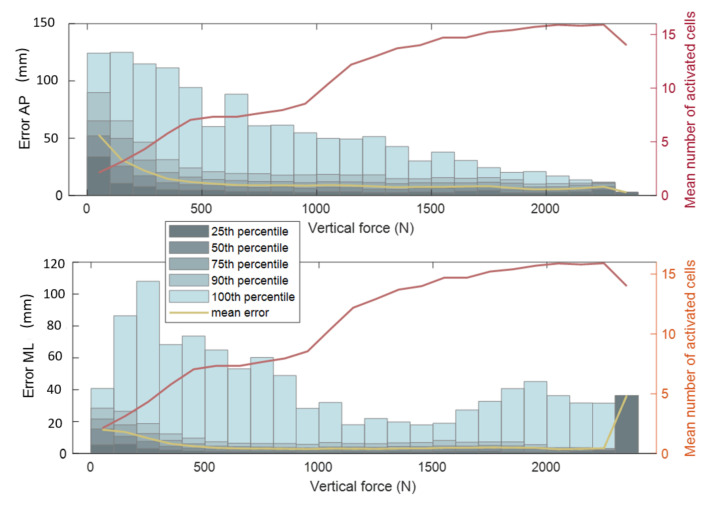
AP and ML component of the absolute error of piCoP as a function of the vertical force measured by the pressure insoles. The vertical force were grouped to fit 100 N force intervals. The mean absolute error piCoP is represented for each vertical force interval (yellow). The mean number of activated pressure cell is represented for each force interval (red) on the right axis. The number of data points (N) for each box is indicated at the top of the graph window.

## Data Availability

Not applicable.

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
