# Peer review of "Evaluation of the Foot Center of Pressure Estimation from Pressure Insoles during Sidestep Cuts, Runs and Walks"

_sensors, 2022, doi:10.3390/s22155628_

Round 1
Reviewer 1 Report
The paper assess the possibility to calculate the COP using pressure insoles. The results obtained with this methodology are then compared with the ones obtained by using a golden standard, i.e. force platforms.
The paper is methodologically sound, experiments are planned with attention and the introduction provides a useful glimpse at literature.
Authors should had an evaluation about the accuracy of the estimation of the reference frame of the insoles with respect the force platform reference frames. Without this estimation, the this affects the accuracy of subsequent measurements. Furthermore, the differences found between the position of piCOP and fpCOP could simply be due to an incorrect identification of the reference system.
I have some remarks about the measurement units adopted in the figures. Differences of the estimated piCOM with respect fpCOM should be reported in (mm) instead of (m).
Author Response
Dear reviewer 1,
Thank you for your comments that helped us to improve the quality of the article. Changes in the article have been marked using red writing color.
The paper assess the possibility to calculate the COP using pressure insoles. The results obtained with this methodology are then compared with the ones obtained by using a golden standard, i.e. force platforms.
The paper is methodologically sound, experiments are planned with attention and the introduction provides a useful glimpse at literature.
Authors should had an evaluation about the accuracy of the estimation of the reference frame of the insoles with respect the force platform reference frames. Without this estimation, the this affects the accuracy of subsequent measurements. Furthermore, the differences found between the position of piCOP and fpCOP could simply be due to an incorrect identification of the reference system.
The accuracy of the piCoP positions depends on the accuracy of measurement of the insoles and the accuracy of the estimation of the related reference frame. We minimize the error of the estimation of the reference frame of the insoles by using as calibration trial the trial under study (without totally compensate it). We presented distinct results for various calibration trials. The accuracy difference is related to the estimation of the reference frame of the insoles. We added two sentences in the discussion to underline this limitation.
“The accuracy of the piCoP positions depends on the accuracy of measurement of the insoles and the accuracy of the estimation of the related reference frame. The error of the estimation of the reference frame of the insoles is minimized by using as calibration trial the trial under study but it is not totally compensated.”
I have some remarks about the measurement units adopted in the figures. Differences of the estimated piCOM with respect fpCOM should be reported in (mm) instead of (m).
Thank you for your vigilance on this point. We changed the units accordingly.
Reviewer 2 Report
The article is interesting, but requires some corrections.
In the Introduction, please explain the practical significance of this type of research.
Make a Hypothesis, determine the dependent and independent variables.
As for the Discussions, see the article:
WilczyÅ„ski J, Bieniek K, Margiel K, Sobolewski P, WilczyÅ„ski I, ZieliÅ„ski R. Canonical correlations between postural stability and body posture defects in children. Medical Studies 2022; 38 (1): 6–13. doi.org/10.5114/ms.2022.115142.
Please describe the strengths and weaknesses of the research carried out.
Author Response
Dear reviewer 2,
Thank you for your comments that helped us to improve the quality of the article. Changes in the article have been marked using red writing color.
The article is interesting, but requires some corrections.
In the Introduction, please explain the practical significance of this type of research.
As for the Discussions, see the article: WilczyÅ„ski J, Bieniek K, Margiel K, Sobolewski P, WilczyÅ„ski I, ZieliÅ„ski R. Canonical correlations between postural stability and body posture defects in children. Medical Studies 2022; 38 (1): 6–13. doi.org/10.5114/ms.2022.115142.
We added the reference to an already existing sentence to underline the practical significance of our work.
“The evaluation of CoP position assessment from pressure insole data (piCoP) opens the door to applications where the magnitude of the studied phenomenon is greater than the accuracy determined.”
Please describe the strengths and weaknesses of the research carried out.
We developed the strengths and the weaknesses of our work in the discussion under the section “perspectives of use” and the section “limitations and perspectives”. We added a sentence in the conclusion to underlines key points.
“This study evaluated an upper limit of the pressure insoles accuracy in the pressure
insole coordinate system. This accuracy depends on the calibration trial used for the
localization of the insoles in the reference coordinate system. The most accurate results
were obtained when the motion performed during the calibration trial was similar to the
motion under study.”
Make a Hypothesis, determine the dependent and independent variables.
We modified a sentence of the introduction to clearly identify the dependent and the independent variables.
“The accuracy of the piCoP was studied as a dependent variable of the kind of the motion performed during the calibration trial, the vertical external forces estimated by pressure insoles and the number of pressure cells involved in the piCoP computation considered as independent variables.”
Reviewer 3 Report
Dear author
This is a meaningful study, and it is a well-written document overall.
I'm asking for minor revisions for publication.
Abstract
Please, match decimal places throughout the document, including abstracts.
Please make your conclusion a little more specific and clear.
Introduction
Line 17, … Instead, change it to etc.
Although the introduction describes various information, the overall flow of sentences and paragraphs is complicated. Please edit it to make it more readable.
Purpose of the study: I hope that this study will highlight how it differs from other studies.
In line 32, you wrote “demonstrated lower validity compared to Pedar-X® insoles”, is there a reason to conduct a study using Moticon® insoles?
Is low validity a good thing? Otherwise, change it to “demonstrated reliability and validation”
Line 50 “One calibration trial was used to perform this localization.” You can delete it.
Conclusion
I hope that some more content will be added based on the results.
Author Response
Dear reviewer 3,
Thank you for your comments that helped us to improve the quality of the article. Changes in the article have been marked using red writing color.
This is a meaningful study, and it is a well-written document overall.
I'm asking for minor revisions for publication.
Abstract
Please, match decimal places throughout the document, including abstracts.
Please make your conclusion a little more specific and clear.
Thank you for your vigilance on this point. We changed the decimal places accordingly. We added two sentences to the conclusion to underline the strengths and the weakness of our study.
Introduction
Line 17, … Instead, change it to etc.
Although the introduction describes various information, the overall flow of sentences and paragraphs is complicated. Please edit it to make it more readable.
Purpose of the study: I hope that this study will highlight how it differs from other studies.
In line 32, you wrote “demonstrated lower validity compared to Pedar-X® insoles”, is there a reason to conduct a study using Moticon® insoles?
Is low validity a good thing? Otherwise, change it to “demonstrated reliability and validation”
Line 50 “One calibration trial was used to perform this localization.” You can delete it.
We have taken your writing advice into account.
We modified a sentence of the introduction to clearly identify the dependent and the independent variables. In our study the accuracy of the piCoP was studied as a dependent variable of the kind of the motion performed during the calibration trial. This hypothesis differs from other studies.
Moticon® insoles are one of the pressure insoles sold on the market. These pressure insoles have advantages and weaknesses that we tried to highlight as the lower validity than for Pedar-X® insoles and the high reliability. These remarks seemed important to us in order to analyze and compare our results with those of other studies.
Conclusion
I hope that some more content will be added based on the results.
Further studies are being conducted to add content based on this data.